# Biologic Evaluation of a Heterodimeric HER2-Albumin Targeted Affibody Molecule Produced by Chemo-Enzymatic Peptide Synthesis

**DOI:** 10.3390/pharmaceutics14112519

**Published:** 2022-11-19

**Authors:** Yongsheng Liu, Rezan Güler, Yunqi Liao, Anzhelika Vorobyeva, Olof Widmark, Theodorus J. Meuleman, Anna Koijen, Leendert J. van den Bos, Robert Naasz, Vitalina Bodenko, Anna Orlova, Caroline Ekblad, Vladimir Tolmachev, Fredrik Y. Frejd

**Affiliations:** 1Department of Immunology, Genetics and Pathology, Uppsala University, 751 85 Uppsala, Sweden; 2Affibody AB, 171 65 Solna, Sweden; 3EnzyTag BV, Daelderweg 9, NL-6361 HK Nuth, The Netherlands; 4Research Centrum for Oncotheranostics, Research School of Chemistry and Applied Biomedical Sciences, Tomsk Polytechnic University, 634050 Tomsk, Russia; 5Department of Medicinal Chemistry, Uppsala University, 751 23 Uppsala, Sweden; 6Science for Life Laboratory, Uppsala University, 752 37 Uppsala, Sweden

**Keywords:** Affibody molecule, albumin binding domain (ABD), peptide synthesis, enzymatic ligation, scaffold protein, radionuclide therapy, ^177^Lu, DOTA, SKOV-3 xenograft, biodistribution

## Abstract

Targeted molecular radiation therapy is a promising emerging treatment modality in oncology, and peptide synthesis may shorten the time to reach the clinical stage. In this study, we have explored Chemo-Enzymatic Peptide Synthesis, or CEPS, as a new means of producing a therapeutic HER2 targeted Affibody^®^ molecule, comprising a C-terminal albumin binding domain (ABD) for half-life extension and a total length of 108 amino acids. In addition, a DOTA moiety could be incorporated at N-terminus directly during the synthesis step and subsequently utilized for site-specific radiolabeling with the therapeutic radionuclide ^177^Lu. Retained thermodynamic stability as well as retained binding to both HER2 and albumin was verified. Furthermore, HER2 binding specificity of the radiolabeled Affibody molecule was confirmed by an in vitro saturation assay showing a significantly higher cell-bound activity of SKOV-3 (high HER2 expression) compared with BxPC3 (low HER2 expression), both in the presence and absence of HSA. In vivo evaluation in mice bearing HER2 expressing xenografts also showed specific tumor targeting as well as extended time in circulation and reduced kidney uptake compared with a HER2 targeted Affibody molecule without the ABD moiety. To conclude, we have demonstrated that CEPS can be used for production of Affibody-fusion molecules with retained in vitro and in vivo functionality.

## 1. Introduction

Targeted treatment directed at a specific receptor overexpressed in certain tumors offers an attractive approach to improve efficacy in cancer treatment. The use of monoclonal antibodies has become a successful basic principle for highly selective targeting of extracellular receptor domains, and a prototypic example is the targeting of HER2 and the approval of trastuzumab for HER2-overexpressing breast cancer [1]. To further potentiate the treatment, antibodies can be derivatized with a potent cytotoxic molecule, allowing enrichment of the toxic molecule in the tumor location e.g., trastuzumab emtansine [2]. Use of an even more potent toxin to form trastuzumab deruxtecan demonstrated opportunity for enhanced treatment benefit in patients failing on multiple HER2-treatments [3] and including patients with lower HER2 expression levels that were previously considered non-eligible for HER2 targeted treatment [4]. Clearly, there is a continued need for potent and targeted anticancer drugs.

One option for anti-cancer cell effect is to use a radionuclide as a cytotoxic payload. This type of therapeutic agents emits ionizing radiation and therefore has a different effector modality, which in principle may allow to overcome resistance mechanisms and result in improved efficacy [5]. In contrast to conventional antibody drug conjugates (ADC), radionuclides decay over time, and it is therefore of very high importance to identify the right molecular format of the targeting agent and to understand its kinetic biodistribution properties. Early access to clinical biodistribution data is key to success and small-scale chemical synthesis of promising targeting agents is an efficient way to accelerate this process.

Affibody molecules are a class of targeting agents that can be produced both recombinantly and by chemical synthesis [6]. They consist of small protein domains (molecular weight 6.5 kDa) from which high-affinity binders can be derived [7]. A HER2-specific Affibody molecule has demonstrated utility to localize HER2 expressing metastatic lesions in patients with breast cancer [8]. Due to their small size, Affibody molecules are filtered through glomerular membranes. For imaging applications, this is a substantial advantage, because this enables rapid reduction in background and obtaining high-contrast images. However, Affibody molecules are reabsorbed in proximal tubuli of kidneys, which is a common problem for most small protein domains [9,10]. The exact molecular mechanism of reabsorption remains unclear, but it is not mediated by the megalin-cubulin system, and conventional methods of the reabsorption reduction are inefficient in the case of Affibody molecules [11,12]. This is a limitation for therapeutic translation of promising targeting agents. One way to reduce the kidney uptake could be association to serum albumin by incorporation of a small albumin binding domain (ABD) [10,13]. The apparent larger size of the resulting complex of the targeting agent with albumin would reduce renal filtration and thus uptake, and albumin is documented to have good tumor exposure, which would allow retained tumor targeting [14].

So far, heterodimers containing an Affibody molecule and ABD were produced recombinantly as fusion proteins [10,13]. The use of peptide synthesis instead of recombinant production offers several advantages, such as well-controlled site-specific incorporation of prosthetic groups, the use of unnatural amino acids to increase protease resistance, a chemical manufacturing facilitating the regulatory path and lower cost of goods for small volume production. Peptide synthesis also allows direct incorporation of chemical entities such as DOTA chelator at the N-terminus of a polypeptide. However, a heterodimer of an Affibody molecule including ABD would be a large polypeptide (>100 aa), which could be a challenge to synthesize at scale. We therefore wanted to investigate if Chemo-Enzymatic Peptide Synthesis (CEPS) could be an option for production of an Affibody molecule that would have both good tumor targeting and low kidney uptake. The method is based on separate synthesis of smaller peptide fragments that are subsequently ligated through a subtilisin derived engineered protein ligase named Omniligase [15,16]. By replacing the active-site serine for a cysteine, the subtilisin derived enzymes gain ligating properties and have lost hydrolytic properties. After several engineering rounds focused on improving enzyme stability, pocket tolerance, and synthesis/hydrolysis ratio, the Omniligase enzyme was generated. The pockets in Omniligase involved in docking of the substrate are deliberately designed to recognize a broad pallet of different amino acids (hydrophobic, bulky, charged, etc.). The pockets most influential in docking include P1–P4 N-terminal of the scissile bond and P1′-P2′ C-terminal of the scissile bond. The smaller peptide fragments involved in the Omniligase mediated ligation are produced via Fmoc solid-phase peptide synthesis (SPPS). The N-terminal fragment needs to be decorated with an oxo-ester. This oxo-ester reacts with the active-site cysteine followed by nucleophilic displacement with the backbone amine of the C-terminal fragment (see Figure 1A) [17,18].

The aim of this study was to test hypotheses that a heterodimer containing a HER2-binding Affibody molecule and ABD can be produced using CEPS, retain a high affinity binding to both HER2 and albumin, remain for an extended time in circulation, and target specifically HER2-expressing xenografts in mice. By screening the Affibody molecule several promising ligation sites were identified in the middle of the molecule. The Affibody molecules are characterized by a constant domain (towards the C-terminus) and a variable domain (towards the N-terminus). To keep the highest degree of freedom, a ligation site located in the constant domain was selected, offering the possibility to rapidly design variants by synthesizing a small library of variable domains and ligating them to the constant domain in a modular fashion. To test the hypothesis, we used CEPS to prepare a HER2-targeting Affibody molecule based on Z_HER2:2891,_ conjugated with the versatile chelator DOTA at the N-terminus and a deimmunized ABD variant at the C-terminus (Figure 1B). The construct was denoted PEP40233. Its binding to HER2 and albumin was investigated using SPR. Furthermore, PEP40233 was labeled with the therapeutic radionuclide ^177^Lu via the DOTA moiety. Specificity and affinity of its binding to human cancer cell lines with different level of HER2 expression was evaluated in vitro. Targeting of human tumor xenografts was evaluated in immunodeficient mice. To estimate the effect of fusion with ABD, biodistribution of [^177^Lu]Lu-PEP40233 was compared with the biodistribution of the Affibody molecule [^177^Lu]Lu-ABY-025, which contains a similar HER2 targeting moiety but no ABD (Figure 1C).

## 2. Materials and Methods

### 2.1. General

Peptiligases, such as Omniligase, are a patented technology by EnzyPep B.V. and can be obtained through EnzyTag B.V. (www.enzytag.com (accessed on 15 November 2022)). The enzyme is produced by recombinant expression in Bacillus subtilis and purified via a C-terminal hexahistidine tag.

No-carrier-added ^177^LuCl_3_ was purchased from Curium Pharma (Stockholm, Sweden). Buffers for labeling were purified from metal contaminations using Chelex 100 resin (Bio-Rad Laboratories, Hercules, CA, USA). The NAP-5 size-exclusion columns were purchased from GE Healthcare. Radioactivity was measured with an automated gamma-spectrometer with a NaI (TI) detector (2480 Wizard, Waltham, MA, USA). The radioactivity distribution on the instant thin-layer chromatography (iTLC) strips was measured using a Storage Phosphor System (CR-35 BIO Plus, Elysia-Raytest, Bietigheim-Bissingen, Germany) and analyzed with AIDA Image Analysis software (Elysia-Raytest, Bietigheim-Bissingen, Germany).

In vitro cell studies were performed using ovarian cancer SKOV-3 cell line with high HER2 expression and pancreatic cancer BxPC3 cell line with low HER2 expression, both obtained from the American Type Culture Collection (ATCC, Manassas, VA, USA). Cell culture medium (Roswell Park Memorial Institute 1640 medium from Sigma-Aldrich, St. Louis, MO, USA), was supplemented with 10% fetal calf serum, 2 mM L-glutamine, 100 IU/mL penicillin, and 100 mg/mL streptomycin. Human Serum Albumin (HSA) was purchased from Sigma-Aldrich (Stockholm, Sweden).

Data on cellular uptake and biodistribution were analyzed by a two-tailed *t* test using GraphPad Prism (version 4.00 for Windows GraphPad Software, San Diego, CA, USA) to determine any significant differences (*p* < 0.05).

### 2.2. Production of PEP40233 and ABY-025

Using CEPS, PEP40233 was produced by the enzymatic ligation of two synthetic peptide fragments using the subtilisin derived enzyme Omniligase. For that purpose, a ligation site at the beginning of the constant fragment was chosen, which in principle allows for the combination of different Affibody molecule specificities with the ABD without a requirement to adjust the ligation conditions. The resulting peptide precursors had a length of 47 and 61 amino acids, respectively. For recognition by the Omniligase it is required that the N-terminal fragment has an oxo-ester moiety at its C-terminus. Therefore, the 47-mer was elongated by a carboxyamidomethyl (Cam) ester, which is the ester derivative of a glycine, followed by a leucine residue. The 47-mer was modified with the DOTA on the N-terminus, which also prevents ligation of the 47-mer ester with its own N-terminus resulting in cyclization or polymerization of the peptide. Therefore, no additional protection group at the N-terminus of the 47-mer was required.

Both peptide precursors were synthesized by Almac (Edinburgh, UK) using standard Fmoc-based solid phase peptide synthesis. The Cam ester was installed using Fmoc-glycolic acid as a building block as described by Nuijens et al. [19]. For the ligation, 0.8 µmol of the C-terminal 61-mer was dissolved in 160 µL of 100 mM Tricine buffer pH 8.4 and 40 µL of Omniligase (5 mg/mL) were added. Subsequently, 0.4 µmol of the N-terminal 47-mer was dissolved in this reaction mixture, the pH was re-adjusted to 8.2, and the reaction was incubated at room temperature. 0.4 µmol of the 47-mer was added a second and third time after 1 h and 2 h of incubation, respectively. The reaction was monitored by LC-MS analysis and upon completion of the reaction the mixture was diluted with 3 mL of Milli-Q water and lyophilized. Once lyophilized, the residue was reconstituted in 1 mL Milli-Q water:MeCN:tert-BuOH (2:1:1, *v*/*v*/*v*) and purified by preparative HPLC using a gradient of 35–40% acetonitrile in water (both with 0.1% trifluoroacetic acid) over 9 min and a XBridge Prep C18 5 µm OBD column (Waters, Milford, MA, USA). After purification the product was again lyophilized.

ABY-025 was produced by solid phase peptide synthesis using the Fmoc/tBu strategy as described previously [20]. ABY-025 lacks the ABD moiety and was used for comparison in the in vivo biodistribution study. The detailed characterization of ABY-025 has been performed earlier. The molecular mass of ABY-025 was confirmed using LC-MS.

### 2.3. Characterization of PEP40233

Characterization of PEP40233 was performed to determine purity, isoelectric point (pI) and thermal stability, prior to performing biodistribution studies. The lyophilized molecule was dissolved in Dulbecco’s phosphate-buffered saline (DPBS) to a concentration of 1 mg/mL. To determine the purity, a sample of PEP40233 was loaded on an RP-UPLC column (Agilent Zorbax 300 SB-C8 RRHD; 1.8 µm, 2.1 × 100 mm column) and eluted by a linear gradient of acetonitrile from 25–50% in 0.1% TFA during 8.3 min at a flow rate of 0.5 mL/min, at 40 ℃. The molecular mass of the construct was determined using mass-spectrometry (API electrospray single quadrupole MSD, Agilent Technologies, Santa Clara, CA, USA). Isoelectric focusing was applied to determine the pI of PEP40233 (Novex™ pH 3-10 IEF Protein Gels, ThermoFisher, Manassas, MA, USA). The reversible folding ability of PEP40233 after heating to 90 ℃ was analyzed by circular dichroism spectroscopy (Jasco J-810 spectropolarimeter, Jasco Scandinavia AB, Mölndal, Sweden).

### 2.4. Binding of PEP40233 to HER2, HSA and MSA

The binding kinetics of PEP40233 to HER2 (rh-HER2-Fc, Sino Biological, Beijing, China), human serum albumin (HSA; Albumedix, Nottingham, UK) and mouse serum albumin (MSA; Sigma-Aldrich, Missouri, USA) were determined using Surface Plasmon Resonance (Biacore 8K, Cytiva, Uppsala, Sweden). For binding analysis to HER2, rh-HER2-Fc (1 or 10 µg/mL) was immobilized to a CM5 chip (Cytiva) using amine coupling chemistry. Single cycle kinetic analysis was performed using HPS-EP+ running buffer (Cytiva). PEP40233 was injected in a dilution series at concentrations of 0.5, 1, 2, 4, and 8 nM. Measurement in the presence of 100 nm HSA or MSA was also performed at a concentration of 8 nM injected PEP40233. For binding analysis to albumin, HSA or MSA (10 µg/mL) were immobilized to a CM5 chip and PEP40233 was injected in a dilution series at concentrations of 0.5, 1, 2, 4 and 8 nM.

### 2.5. Labeling Chemistry

For labeling, an aliquot of 90 µg of PEP40233 in Milli-Q water was mixed with 25 µL of 0.2 M ammonium acetate, pH 6.0. A predetermined amount (15 MBq) of ^177^Lu was added, followed by vortexing. After incubation at 65 °C for 30 min, the radiochemical yield was analyzed by iTLC (~1 µL samples). The iTLC strips were eluted with 0.2 M critic acid, pH 2.0. The labeling of ABY-025 with ^177^Lu was performed as described previously [13].

### 2.6. In Vitro Studies

The binding specificity of radiolabeled PEP40233 to HER2-expressing cells was evaluated using SKOV-3 (1.6 × 10^6^ HER2 receptors per cell) [21] and BxPC3 (1.1 × 10^4^ receptors per cell) [22] cells. Experiments were performed in triplicate. Approximately 10^6^ cells were seeded per well in 6-well plates the day before the experiment and were maintained in 2 mL complete RPMI 1640 medium at 37 °C and 5% CO_2_. A 1000-fold excess of non-labeled PEP40233 was added, followed by incubation at room temperature (RT) for 30 min to saturate the receptors. Thereafter, all cells were incubated with labeled conjugates (0.5 nM) for 60 min at 37 °C. After that, the medium was discarded, and the cells were washed once with 1 mL PBS. Then, the cells were incubated with 1 mL Trypsin-EDTA (Thermo Fisher Scientific, Waltham, MA, USA) at 37 °C to allow detachment. The cells were collected and washed with 1 mL PBS. The radioactivity in cells and standards was measured by an automatic gamma counter to calculate the percentage of cell-bound radioactivity. To evaluate the impact of HSA on binding specificity, HSA was added to the complete RPMI 1640 medium to obtain a concentration of 100 nM and the HSA-containing medium was used for the experiment described.

The affinity of radiolabeled PEP40233 binding to SKOV-3 cells was measured using LigandTracer (Ridgeview Instruments AB, Vänge, Sweden). Cells were seeded on the local area of a cell culture dish (NunclonTM, Size 100620, NUNC A/S, Roskilde, Denmark) one day before the experiment. The radiolabeled PEP40233 was added at concentrations of 0.25 nM, 0.75 nM, and 2.25 nM for each affinity assay. RPMI 1640 medium containing 100 nM HSA was used to investigate the influence of HSA on the binding affinity. The real-time kinetics of binding and dissociation were observed according to the description of Björke and Andersson [23]. The dissociation constant at equilibrium (KD) was analyzed by InteractionMap software (version 1.9.2, Ridgeview Diagnostics AB, Uppsala, Sweden). The data were treated with the InteractionMap following the instruction described by Altschuh and co-workers [24].

### 2.7. In Vivo Studies

The animal experiments were performed in accordance with national legislation on laboratory animal protection, and the study was approved by the local Ethics Committee for Animal Research in Uppsala (permit 5.8.18-11931/2020, 28 August 2020). An overdosing of Rompun/Ketalar anesthesia was used for animal euthanasia.

The biodistribution measurement of ^177^Lu-labeled PEP40233 was performed in female BALB/C nu/nu mice bearing xenografts with different HER2 expression levels. Xenografts with high and low HER2 expression were established by subcutaneous injection of approximately 10^7^ SKOV-3 cells and BxPC3 cells, respectively, on the abdomen of the mice. Four mice were used in each group and the average animal weight was 21.2 ± 1.2 g at the time of the experiment. One group of mice with SKOV-3 xenografts and one group with BxPC3 xenografts were injected intravenously with [^177^Lu]Lu-PEP40233 in 100 µL of PBS. The injected activity was 270 kBq/mouse, and the injected protein was adjusted to 10 µg/mouse using non-labeled PEP40233. The average tumor weight was 0.2 ± 0.2 g. To investigate the effect of ligation with ABD, one group of mice bearing SKOV-3 xenografts was injected intravenously with [^177^Lu]Lu-ABY-025 (HER2 targeting Affibody molecule without ABD) at an injected activity of 260 kBq/mouse with the protein dose of 5 µg/mouse.

The blood, heart, lung, liver, spleen, pancreas, kidneys, tumor, muscle, bone, gastro-intestinal tract, and remaining carcass were collected, and the biodistribution was measured at 48 h after injection. Organs and tumor uptake were calculated as the percentage of the injected dose per gram of the sample (% ID/g), while for intestines and carcass uptake, the percentage of injected dose (% ID) per the whole sample was calculated.

In vivo imaging was performed to obtain an additional verification of the biodistribution data. One mouse with SKOV-3 xenograft was injected i.v. with 1.7 MBq (10 µg) of [^177^Lu]Lu-PEP40233 48 h before the imaging, which was performed using nanoScan SPECT/CT (Mediso Medical Imaging Systems, Budapest, Hungary). The animals were anesthetized using sevoflurane and placed in a prone position on a warmed scanner bed. Anesthesia was maintained throughout the study using 1.5–2.0% sevoflurane in 50% oxygen and 50% medical air (flow of 0.5 L/min). The imaging protocol was the same as described by Liu and co-workers (2021).

## 3. Results

### 3.1. Production and Characterization of PEP40233

PEP40233 was produced by the Omniligase catalyzed ligation of a C-terminal fragment of 61 amino acids to an N-terminal fragment of 47 amino acids, which was elongated by a Cam-ester at its C-terminus. As the separation of the 61-mer from the ligation product by preparative HPLC was difficult, full conversion of the 61-mer was ensured by using a 1.5-fold excess of the 47-mer ester during the ligation. A possible side reaction of the ligation is hydrolysis of the ester moiety. In order to minimize hydrolysis of the Cam-ester, the 47-mer ester fragment was added to the ligation reaction in three portions. Upon full conversion of the peptide fragments, the ligation product was purified via preparative RP-HPLC. Ligations were performed on a 0.8 µmol scale and on average yielded 3.8 mg of product, which corresponds to a yield of 39% for the ligation and purification. The correct molecular weight of PEP40233 (calculated Mw: 12,278.9 Da, found Mw: 12,280.3 Da) was confirmed by RP-UPLC-MS analysis and the purity of the synthesized sample was >95% (Appendix A). In the MS analysis, PEP40233 showed significant coordination with various salts (i.e., K+ and Na+), which likely can be attributed to the chelating properties of the DOTA group. The pI of PEP40233 was determined to be 5.2 by isoelectric focusing (Appendix A). Measurement by circular dichroism spectroscopy showed that PEP40233 had a typical alpha helical structure. The CD spectra measured at 20 °C before and after heating to 90 °C overlapped (Figure 2), confirming reversible refolding after heat denaturation.

### 3.2. Affinity Measurements Using SPR

PEP40233 produced by CEPS showed retained high-affinity binding to HER2 and albumin (Figure 3). The binding affinities to HER2, HSA, and MSA determined by single cycle kinetic analysis are shown in Table 1. Measurements in the presence of 100 nM of MSA or HSA reduced the affinity of PEP40233 to HER2 by approximately one order of magnitude.

### 3.3. Labeling Chemistry

The labeling of PEP40233 was rapid and efficient, providing a radiochemical yield of 98.5% after 30 min of incubation at 65 °C. No additional purification was needed due to the high yield. The radiochemical yield of labeling ABY-025 was also over 98%.

### 3.4. In Vitro Studies

The result of the in vitro binding specificity of [^177^Lu]Lu-PEP40233 is shown in Figure 4. The binding of [^177^Lu]Lu-PEP40233 to SKOV-3 and BxPC3 cells after saturation of HER2 receptors with unlabeled PEP40233 was significantly (*p* < 0.05) lower both in the presence and absence of HSA. The cell-associated activity of SKOV-3 was significantly (*p* < 0.05) higher than that of BxPC3, both in the presence and absence of HSA. When comparing the cell-bound activity, no significant reduction (*p* > 0.05) was observed in the presence of HSA.

The affinity was evaluated by InteractionMap analysis of the LigandTracer sensorgrams. The data are presented in Table 2 and in Figure 5. The binding of ^177^Lu-labeled PEP40233 to HER2-expressing SKOV-3 cells in the presence of HSA showed two kinds of interactions one with a higher sub-nanomolar affinity (36%) and one with a lower nanomolar affinity (44%).

### 3.5. In Vivo Studies

Data concerning the biodistribution of [^177^Lu]Lu-PEP40233 in BALB/C nu/nu mice bearing high HER2-expressing SKOV-3 xenografts and low HER2-expressing BxPC3 xenografts at 48 h after injection are shown in Appendix A and Figure 6A. In mice bearing SKOV-3 xenografts, the uptake of radioactivity in kidney was the highest compared with that in other normal organs and the tumor uptake was 1.6-fold higher (significant difference, *p* < 0.05) than the renal uptake. In mice bearing low HER2-expressing BxPC3 xenografts, the tumor uptake was significantly (*p* < 0.05) lower than the renal uptake. A comparison showed a significantly higher (*p* < 0.05) tumor uptake in SKOV-3 xenografts compared with BxPC3 xenografts. Interestingly, the renal uptake in mice with SKOV-3 xenografts was significantly lower than in mice bearing BxPC3 xenografts.

A comparison with [^177^Lu]Lu-ABY-025, which lacks the albumin binding domain, in SKOV-3 xenograft-bearing mice at 48 h after injection showed that the renal uptake of [^177^Lu]Lu-PEP40233 was 9.4-fold lower (*p* < 5 × 10^−5^) than the renal uptake of [^177^Lu]Lu-ABY-025. Furthermore, the blood concentration of [^177^Lu]Lu-PEP40233 was 265-fold higher (*p* < 5 × 10^−4^) than the blood concentration of ^177^Lu-ABY-025. The increased retention in [^177^Lu]Lu-PEP40233 in blood resulted in significantly (*p* < 0.05) higher uptake in heart muscle, lung, liver, spleen, pancreas, muscle, and bone in comparison with [^177^Lu]Lu-ABY-025. However, there was no significant (*p* > 0.05) difference in tumor uptake of [^177^Lu]Lu-PEP40233 and [^177^Lu]Lu-ABY-025 (Figure 6B). The uptake of [^177^Lu]Lu-PEP40233 in the whole gastrointestinal tract (including content) was 3.0 ± 0.6 and 2.8 ± 0.2 %ID for animals bearing SKOV-3 and BxPC3, respectively (no significant difference, *p* > 0.05, between animals with different xenograft models). This was significantly (*p* < 0.0001, one-way ANOVA with Dunnett’s multiple comparisons test) higher than the uptake of [^177^Lu]Lu-ABY-025 (0.38 ± 0.05% ID).

An imaging of SKOV-3 xenograft-bearing mouse performed 48 h after injection of [^177^Lu]Lu-PEP40233 (Figure 7) was in agreement with the biodistribution data measured ex vivo. The highest uptake was observed in the tumor. Other organs with detectable uptake were kidneys, but the renal uptake was noticeably lower than the tumor uptake. Accumulation of activity in other organs and tissues was 5–10-fold lower than in the tumor.

## 4. Discussion

Engineered scaffold proteins (ESPs) are an emerging type of targeting agents in oncology. This type offers potential advantages of stability, low-cost production, and easy modification of the design of such targeting agents. Preclinical and early clinical investigation have demonstrated clear advantages of ESP-based imaging agents in comparison with analogues based on monoclonal antibodies [25,26]. This advantage is determined first and foremost by a combination of the high affinity and the small size of the ESP-based probes for molecular imaging. Particularly, this size enables rapid clearance on unbound agent from blood providing a high contrast of imaging. Nevertheless, the small size and rapid clearance of ESP are obstacles for their therapeutic application. These factors reduce bioavailability of targeting probes and might result in accumulation of cytotoxic payloads in kidneys. To overcome this problem for radionuclide therapy, three approaches have been developed: (1) the use of pre-targeting; (2) the use of non-residualizing radioactive labels; (3) increasing the size to avoid glomerular filtration [26]. The first two approaches are specific for the use of radionuclides as cytotoxic substances. The increase in size has more general applications and can be utilized not only for targeted radionuclide therapy [10], but also for the delivery of drugs (see e.g., Xu et al., 2020 and Brandl et al., 2020 [27,28]) or the inhibition of proliferation-driving signaling [29,30]. The size increase effect might be obtained by the addition of large unstructured polypeptides [28] or by adding another moiety binding to the host’s serum albumin [29,30]. Any of these approaches increase the length of therapeutic ESP-based constructs.

Chemical synthesis of therapeutic candidates may shorten the route to clinical evaluation. Short polypeptides are routinely synthesized, whereas challenges to provide sufficient yields increase with peptide length. One approach for enabling synthesis also of longer polypeptides is CEPS, which besides the potential to increase the yield provides flexibility in the production of similar therapeutic candidates for pre-clinical evaluation. Peptide synthesis of shorter fragments offers the possibility to combine conserved fragments that may be produced in larger quantities for off-the shelf use, with variable fragments that may be produced in smaller quantities for specific purposes. Thereby a panel of molecules with different properties may be provided within relatively short time once a bulk of constant fragments are available. In this study, we evaluated the in vitro and in vivo properties of a CEPS produced HER2 targeted Affibody molecule fused to ABD (108 a.a. total length). A DOTA chelator for subsequent radiolabeling was incorporated at the N-terminus of the molecule directly during the synthesis step. This eliminated the need for introducing a cysteine residue and for performing a subsequent conjugation that would have been required if the molecule had been produced recombinantly. Moreover, a constant C-terminal domain applicable to any Affibody construct could in principle be GMP-produced and ready on the shelf for subsequent conjugation to the variable domains of a new Affibody molecule, dramatically shortening the time and cost to clinical studies. The precondition for this is that the CEPS-produced Affibody molecules (and other ESP-based therapeutic conjugates) retain both their capacity to bind to their molecular targets and remain longer in circulation.

The use of CEPS provided a polypeptide with an expected mass and high purity (Appendix A). The expected high helical content of the protein was confirmed (Figure 2). Moreover, the CD measurements at variable temperature demonstrated that the high helical content was preserved after heating up to 90 ℃, which suggest a high-fidelity refolding after heating to temperatures exceeding temperatures that are requited for labelling using macrocyclic chelators. The CEPS produced PEP40233 showed retained binding to both HER2 and albumin (Figure 3 and Table 1). The HER2 binding specificity of radiolabeled PEP40233 was confirmed by an in vitro saturation assay (Figure 4). A significantly higher cell bound activity of SKOV-3 (high HER2 expression) compared with BxPC3 (low HER2 expression) was observed both in the presence and absence of HSA. This provided additional evidence showing that binding of [^177^Lu]Lu-PEP40233 to cancer cells in vitro is HER2-mediated. Furthermore, affinity of [^177^Lu]Lu-PEP40233 binding to living HER2-expressing cells demonstrated binding sites with higher (subnanomolar) and lower (low nanomolar) affinities (Figure 5 and Table 2). Presence of two interactions on HER2 with different affinities is typical for many tracers and may be caused by dimerization level and hence the conformational status of HER2 on the surface of cells [31]. Importantly, even the strength of low-affinity interaction (15.2 ± 9.0 nM) is sufficient for successful tumor targeting using a targeting protein with slow clearance from the blood [32].

In vivo, the tumor uptake of [^177^Lu]Lu-PEP40233 was significantly higher in SKOV-3 xenograft-bearing mice at 48 h after injection compared with the uptake in BxPC3 xenograft-bearing mice (Figure 6A). This demonstrates that uptake of [^177^Lu]Lu-PEP40233 in tumors in vivo depends on HER2 expression level, i.e., is HER2 specific. This confirms our hypothesis that heterodimers of Affibody molecules with ABD, which were produced using CEPS technology, can target specifically HER2-expressing xenografts in mice. Furthermore, a comparison of the biodistribution of [^177^Lu]Lu-PEP40233 with the non-ABD fused [^177^Lu]Lu-ABY-025 (Figure 6B) demonstrated that the heterodimer remains much longer in the blood, confirming our hypothesis. Importantly, the fusion with ABD reduced the renal uptake 9.4-fold. The reduced renal uptake and retained tumor uptake of [^177^Lu]Lu-PEP40233 compared with [^177^Lu]Lu-ABY-025 implies a more favorable tumor-to-kidney ratio of the radiolabeled ABD-ligated PEP40233. The hepatic uptake was significantly higher for [^177^Lu]Lu-PEP40233 compared with [^177^Lu]Lu-ABY-025. The hepatic uptake of [^177^Lu]Lu-PEP40233, 5.5 ± 0.7 %ID/g, was approximately the same as for ^177^Lu-labeled monoclonal anti-HER2 antibody trastuzumab, 5.5 ± 0.7 %ID/g at the same time point [33]. Most likely, it can be explained by longer residence of both [^177^Lu]Lu-PEP40233 and [^177^Lu]-trastuzumab in blood compared with [^177^Lu]Lu-ABY-025. The activity in gastrointestinal tract with content, which might be a measure of hepatobiliary excretion, and was also higher for [^177^Lu]Lu-PEP40233 compared with [^177^Lu]Lu-ABY-025. However, it was approximately equal to the activity of homologous construct [^177^Lu]Lu-ABY-271, which was produced recombinantly [13]. Thus, the enzymatic ligation does not have any impact of hepatic uptake or hepatobiliary excretion of ABD-coupled HER2-targeting Affibody molecules. The tumor uptake of [^177^Lu]Lu-PEP40233, 37 ± 10% ID at 48 h was higher than the uptake of [^177^Lu]- trastuzumab, 14.6 ± 1.8% ID at 48 h when the same tumor model, SKOV3 xenografts in nude mice was used [33].

## 5. Conclusions

A heterodimer, which contains a HER2-binding Affibody molecule and ABD, can be produced using CEPS, and retain a high affinity binding to both HER2 and albumin, remain an extended time in circulation with reduced kidney uptake and target specifically and efficiently HER2-expressing xenografts in mice.

## Figures and Tables

**Figure 1 pharmaceutics-14-02519-f001:**
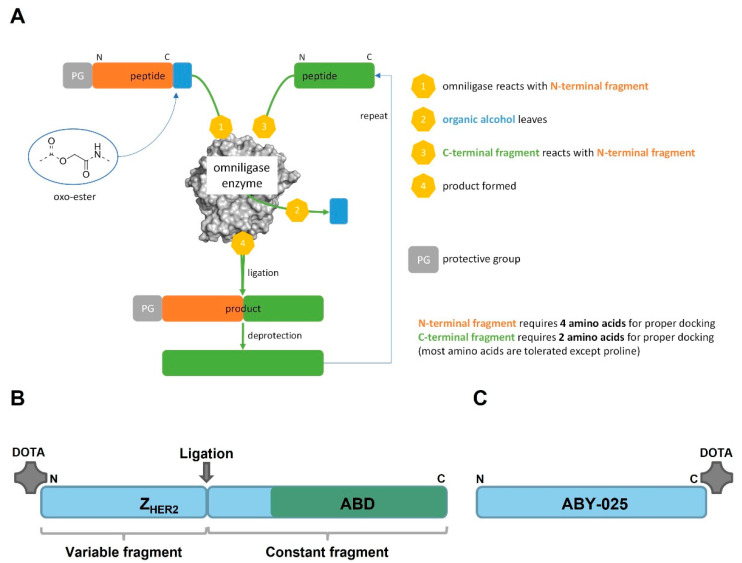
(**A**) Overview of the Chemo-Enzymatic Peptide Synthesis (CEPS) method. Smaller peptide fragments are synthesized separately and subsequently joined by Omniligase. The detailed reaction scheme can be found in Appendix A. (**B**) Schematic of PEP40233 produced by CEPS linking a variable fragment that is interchangeable for different binding specificities (here HER2) and a constant fragment comprising an ABD moiety for half-life extension. A chelating DOTA moiety may optionally be incorporated at the N-terminus during the chemical synthesis. (**C**) Schematic of the HER2 targeted Affibody molecule ABY-025 with a DOTA moiety conjugated at the C-terminus.

**Figure 2 pharmaceutics-14-02519-f002:**
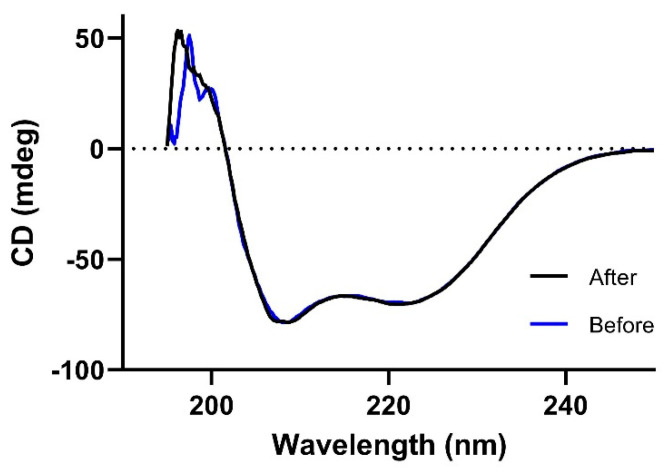
CD spectra recorded at 20 °C before (blue line) and after (black line) heating to 90 °C. The data confirm refolding after heat denaturation.

**Figure 3 pharmaceutics-14-02519-f003:**
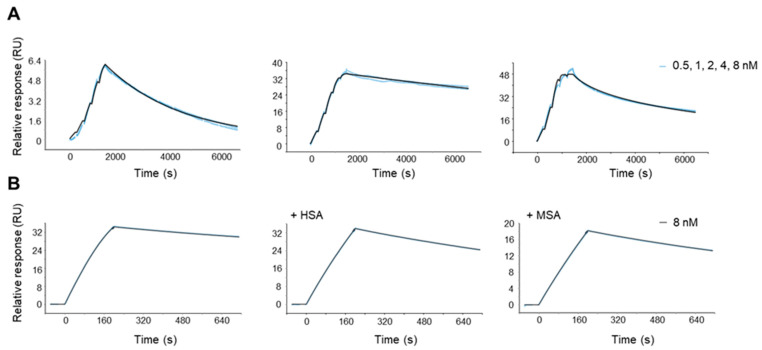
(**A**) Single-cycle kinetics SPR sensorgrams showing PEP40233 concentration series (0.5, 1, 2, 4, 8 nM) interaction curve with, from left to right, immobilized Her2, HSA, and MSA. (**B**) A-B-A injection sensorgrams showing PEP40233 (8 nM) binding to HER2 in the presence of HSA/MSA. The data confirm binding to HER2 both in presence and absence of albumin.

**Figure 4 pharmaceutics-14-02519-f004:**
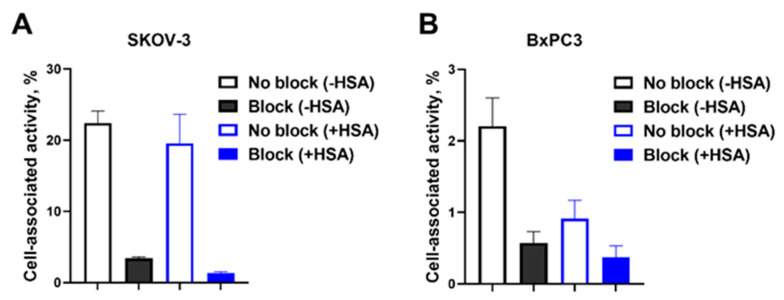
In vitro specificity of [^177^Lu]Lu-PEP40233 binding to (**A**) SKOV-3 and (**B**) BxPC3 cells in the presence and absence of HSA. Block refers to pre-saturation of HER2 receptors with unlabeled PEP40233. The data are presented as an average value from 3 samples ± SD. The data demonstrate specific binding to both cell lines, and both in the presence and absence of HSA.

**Figure 5 pharmaceutics-14-02519-f005:**
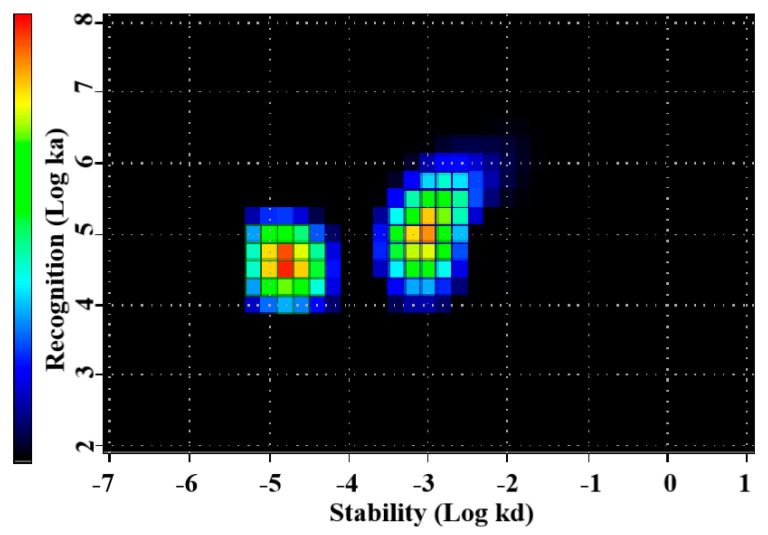
InteractionMap of [^177^Lu]Lu-PEP40233 binding to HER2-expressing SKOV-3 cells in the presence of HSA. Binding was measured at concentrations of 0.25 nM, 0.75 nM, and 2.25 nM. Data are representatives from duplicates. The data show presence of two types of interactions with living HER2-expressing cells.

**Figure 6 pharmaceutics-14-02519-f006:**
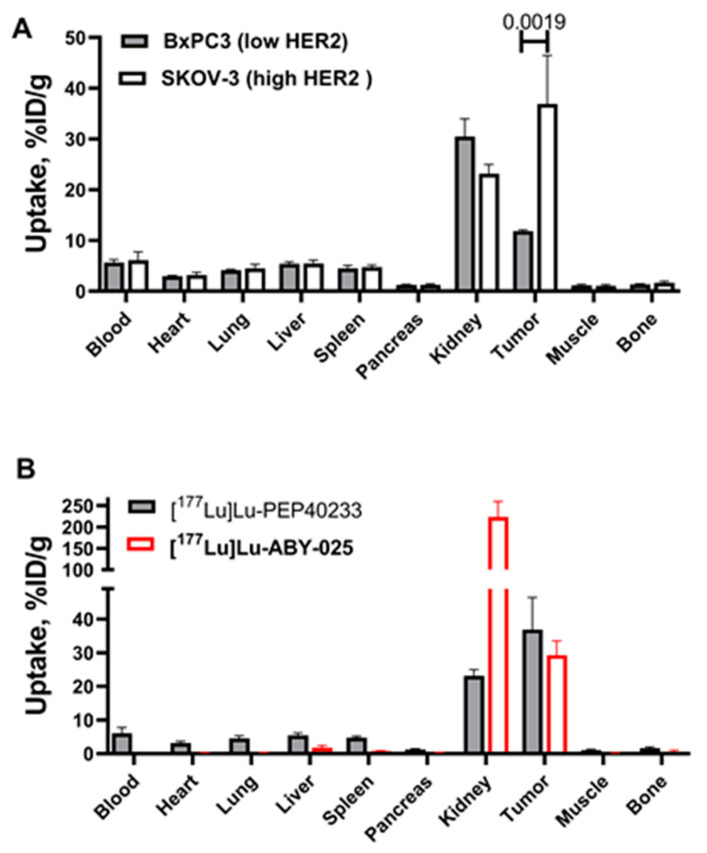
Biodistribution at 48 h post injection. (**A**) Biodistribution of [^177^Lu]Lu-PEP40233 in BALB/C nu/nu mice bearing SKOV-3 and BxPC3 xenografts. Data are presented as an average value from four mice ± SD. The data show higher accumulation in xenografts with higher HER2 expression. (**B**) Comparison of biodistribution of [^177^Lu]Lu-PEP40233 and [^177^Lu]Lu-ABY-025 in SKOV-3 bearing xenografts. Data are presented as an average value from four mice ± SD. The data show clear reduction in the renal uptake and extension of residence in circulation for [^177^Lu]Lu-PEP40233.

**Figure 7 pharmaceutics-14-02519-f007:**
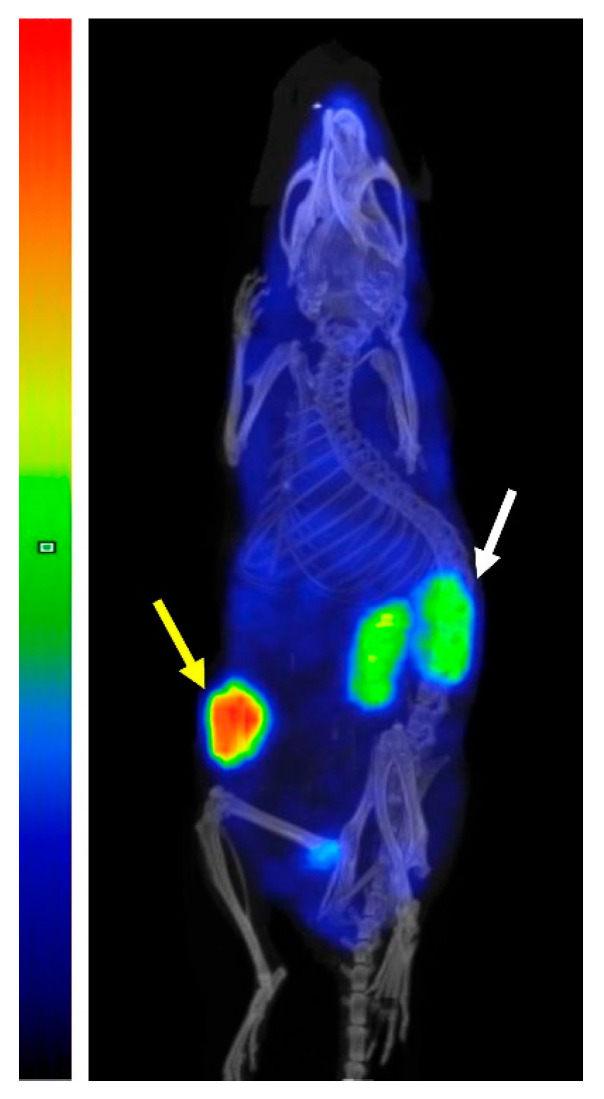
Imaging of mouse bearing SKOV-3 xenograft 48 h after injection of [^177^Lu]Lu-PEP40233 using microSPECT/CT. Tumor (yellow arrow) and kidneys (white arrow) have elevated activity uptake, but tumor uptake exceeds the renal uptake. The scale is linear showing arbitrary units normalized to a maximum count rate.

**Table 1 pharmaceutics-14-02519-t001:** Kinetic parameters of PEP40233 interacting with HER2, HSA and MSA.

Target	k_a_ (1/Ms)	k_d_ (1/s)	K_D_ (M)
HSA	1.58 × 10^6^	4.71 × 10^−5^	2.99 × 10^−11^
MSA	9.04 × 10^6^	4.71 × 10^−4^	5.20 × 10^−11^
HER2	2.06 × 10^6^	2.82 × 10^−4^	1.44 × 10^−10^
HER2 in the presence of 100 nM HSA	1.23 × 10^5^	6.53 × 10^−4^	5.30 × 10^−9^
HER2 in the presence of 100 nM MSA	1.01 × 10^5^	6.47 × 10^−4^	6.42 × 10^−9^

**Table 2 pharmaceutics-14-02519-t002:** InteractionMap evaluation of the affinity of ^177^Lu labeled PEP40233 binding to HER2-expressing SKOV-3 cells in the presence of HSA.

Interaction	k_a_ (M^−1^s^−1^)	k_d_ (s^−1^)	K_D_ (M)	Weight (%)
1	[2.8 ± 1.4] × 10^4^	[17.3 ± 1.9] × 10^−6^	[743 ± 448] × 10^−12^	36
2	[8.0 ± 4.3] × 10^4^	[10.3 ± 0.6] × 10^−4^	[15.2 ± 9.0] × 10^−9^	44

## Data Availability

Data is contained within the article.

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
