# Peer review of "Biologic Evaluation of a Heterodimeric HER2-Albumin Targeted Affibody Molecule Produced by Chemo-Enzymatic Peptide Synthesis"

_pharmaceutics, 2022, doi:10.3390/pharmaceutics14112519_

Round 1

Reviewer 1 Report

The authors need to give information on hepatic clearance of their system

Author Response

Comment

The authors need to give information on hepatic clearance of their system

Answer. Thank you for pointing at this! This is an essential information indeed. To address this comment, we have added the Supplemental Table S1 information concerning the uptake of [177Lu]Lu-PEP40233 is whole GI tract (as %ID per whole sample). We have added also the following sentences to the Results section, In vivo studies subsection The uptake of [177Lu]Lu-PEP40233 in the whole gastrointestinal tract (including content) was 3.0 ± 0.6 and 2.8 ± 0.2 %ID for animals bearing SKOV-3 and BxPC3, respectively (no significant difference, p > 0.05, between animals with different xenograft models). This was significantly (p < 0.0001, one-way ANOVA with Dunnett's multiple comparisons test) higher than the uptake of [177Lu]Lu-ABY-025 (0.38 ± 0.05% ID).“ We have also added to the Discussion section the following text: “The hepatic uptake was significantly higher for [177Lu]Lu-PEP40233 compared to [177Lu]Lu-ABY-025. The hepatic uptake of [177Lu]Lu-PEP40233, 5.5 ± 0.7 %ID/g, was approximately the same as for 177Lu-labeled monoclonal anti-HER2 antibody trastuzumab, 5.5 ± 0.7 %ID/g at the same time point [33]. Most likely, it can be explained by longer residence of both [177Lu]Lu-PEP40233 and [177Lu]-trastuzumab in blood compared to [177Lu]Lu-ABY-025. The activity in gastrointestinal tract with content, which might be a measure of hepatobiliary excretion, was also higher for [177Lu]Lu-PEP40233 compared to [177Lu]Lu-ABY-025. However, it was approximately equal to the activity of homologous construct [177Lu]Lu-ABY-271, which was produced recombinantly [13]. Thus, the enzymatic ligation does not have any impact of hepatic uptake or hepatobiliary excretion of ABD-coupled HER2-targeting affibody molecules.   “

Reviewer 2 Report

Manuscript entitled Biologic evaluation of a heterodimeric HER2-albumin targeted Affibody molecule produced by Chemo-Enzymatic Peptide Synthesis is a well writen article. The topic has a great importance in the field of the diagnosis of HER2 positive tumors.  

Based on the manuscript, reviewer is recommending publication in Pharmaceutics after a minor revision. The main reasons for this recommendation are the following: (i) For the benefit of the chemist reader, relevant chemical formulas and the reaction scheme for preparation of the peptide conjugates should be presented explicitly. It would be important to determine what kind of side-reactions should be occured. (ii) The chemical characterization of ABY-025 is missing, only PEP40233 is presented in the Supplementary file (or not available for the reviewer). (iii). The Figure legends do not contain their brief interpretation.

Author Response

Answer> Thank you for appreciation of our work

Based on the manuscript, reviewer is recommending publication in Pharmaceutics after a minor revision. The main reasons for this recommendation are the following:

(i) For the benefit of the chemist reader, relevant chemical formulas and the reaction scheme for preparation of the peptide conjugates should be presented explicitly. It would be important to determine what kind of side-reactions should be occured.

Answer: Thank you for suggesting a more thorough information for the benefit of the chemist reader. We have added tan additional figure S1 in the supplemental figures describing the chemical reactions (please find uploaded , and a text clarifying the reactions. Full analysis of the detailed reaction is described in references to this technology within the manuscript as that is outside the scope of the current work.  

Figure S1. Reaction scheme for the omniligase catalyzed ligation of two peptide fragments. The N-terminal fragment is modified with an ester-moiety (blue)at its C-terminus. The ligase reacts with the ester, forming a covalent thio-ester intermediate, and the organic alcohol leaves the active site. Subsequently a peptide bond is formed by nucleophilic displacement with the backbone amine of the C-terminal fragment. A possible side reaction is hydrolysis of the thio-ester.

(ii) The chemical characterization of ABY-025 is missing, only PEP40233 is presented in the Supplementary file (or not available for the reviewer).

Answer: Thank you for pointing us to the fact of not full description of ABY-025. In fact, ABY-025 has been thoroughly characterized as it is a product used in clinical research, and the full reference (no 20) is describing the chemical characterization. We have added the following text in the manuscript: “The detailed characterization of ABY-025 has been performed earlier. The molecular mass of ABY-025 was confirmed using LC-MS.“

 (iii). The Figure legends do not contain their brief interpretation.

Answer. To address this comment, we have added brief interpretation (a single sentence) to the legends of Figures.